# Biosynthesis of Bonelike Apatite 2D Nanoplate Structures Using Fenugreek Seed Extract

**DOI:** 10.3390/nano10050919

**Published:** 2020-05-09

**Authors:** Abdalla Abdal-hay, H. Fouad, Basheer A. ALshammari, Khalil Abdelrazek Khalil

**Affiliations:** 1School of Dentistry, University of Queensland, Herston Campus, St Lucia, Queensland 4072, Australia; 2Department of Mechanical Engineering, Faculty of Engineering, South Valley University, Qena 83523, Egypt; 3Applied Medical Science Department, Community College, King Saud University, P.O Box 10219, Riyadh 11433, Saudi Arabia; menhfef@ksu.edu.sa; 4Biomedical Engineering Department, Faculty of Engineering, Helwan University, Helwan 11792, Egypt; 5Materials Science Research Institute, King Abdulaziz City for Science and Technology, P.O Box 6086, Riyadh 11442, Saudi Arabia; bshammari@kacst.edu.sa; 6Department of Mechanical and Nuclear Engineering, College of Engineering, University of Sharjah, Sharjah 27272, UAE

**Keywords:** bone tissue engineering, PCL, biodegradable composite materials, hydroxyapatite

## Abstract

An innovative, biomimetic, green synthesis approach was exploited for the synthesis of humane and environmental friendly nanomaterials for biomedical applications. Ultrafine bonelike apatite (BAp) 2D plate-like structures were prepared using fenugreek seed extract during the biosynthesis wet-chemical precipitation route. The chemical analysis, morphology and structure of the prepared 2D nanoplates were characterized by inductively coupled plasma atomic emission spectroscopy (ICP-OES), electron microscopy (SEM and TEM), X-ray diffraction (XRD) and Fourier transform infrared (FTIR) spectroscopy. A 2D plate-like nanostructure of BAp with an average width (length) of 12.67 ± 2 nm and thickness of 3.8 ± 1.2 nm was obtained. BAp 2D crystals were tuned by interaction with the fenugreek organic molecules during the fabrication process. In addition to Ca and P ions, bone mineral sources such as K, Mg, Na, SO_4_ and CO_3_ ions were incorporated into BAp nanoplates using fenugreek seed extract. The overall organic molecule concentration in the reaction process increased the effectiveness of hydroxyl groups as nucleation sites for BAp crystals. Accordingly, the size of the biosynthesized BAp plate-like structure was reduced to its lowest value. Biosynthesis BAp 2D plate-like nanocrystals showed good viability and higher growth of MC3T3 osteoblast-like structures than that of the control sample. BAp 2D nanoplates prepared by a facile, ecofriendly and cost-effective approach could be considered a favorable osteoconductive inorganic biomaterial for bone regeneration applications.

## 1. Introduction

Biological apatite (non-stoichiometric hydroxyapatite) (BAp) is the main inorganic constituent in human bones and teeth [1]. Pure hydroxyapatite (HAp, Ca_10_(PO_4_)_6_(OH)_2_) has inorganic components of Ca and P ions; however, biological apatite is amorphous and contains several other ions, such as carbonate, Mg^2+^, SO_4_^3−^, Na^+^, CO_3_^2−^, K^+^ and Cl^−^, among others [2]. In addition, the 2D plate-like structure is the predominant phase in biological apatite, due to its intrinsic tendency to grow in a plate-like structure under physiological conditions [2,3]. The fabrication of BAp with a similar chemical composition to natural bone with a controlled 2D plate-like structure is extremely difficult, especially when the mineralization occurs in a complex bioinspired condition. Designing BAp 2D nanoplates is a crucial factor for the improvement of its biological properties [2,4,5]. It has been reported that the composition, size, shape and controlled structure of bioactive bone minerals play a crucial role in determining the physical and chemical properties enabling its biomedical applications [5]. Although there have been many attempts to synthesize BAp minerals with nanoplate morphology, the proposed methods are replete with several problems, including the use of a toxic template and an organic solvent, and creating hazardous by-products [5,6,7], and it is difficult to obtain a 2D plate-like structure simulating the morphology of natural bone [3,8,9]. To drive homogeneous nucleation, a very high degree of supersaturation is required. This is dependent on the crystal-like arrangement and formation of ions meeting the thermodynamic criteria of the critical dimensions. Therefore, the synthesis of BAp nanocrystals using efficient green chemistry routes can be an alternative method of incorporating other biological minerals into precipitated bone minerals. Green synthesis has been intensively employed in recent studies because it is simple, biocompatible, eco-friendly and cost-effective [5,10]. Biomimetic synthesis of its outstanding properties, including the absence of toxic chemicals, high pressure, temperature and/or energy, and its ability to easily scale-up for the large-scale synthesis of nanomaterials [9]. Therefore, the ecofriendly synthesis method is employed in the present study to incorporate bone minerals and control the growth rate of crystal size. Polyphenolic provenance from plants is comprehensively employed in the synthesis of inorganic nanomaterials as stabilizing, reducing and chelating factors [11,12,13,14,15,16], and as eco-friendly alternatives to chemical and physical routes. Fenugreek (FG) is an annual plant belonging to the legume family, which has been used as one of the most promising traditional medicinal herbs. It has been widely used for medicinal purposes such as remedying pain, anti-cancer effects, anti-diabetic medication, anti-microbial effects and as an anti-oxidant reagent [17,18,19]. This is due to the pharmacological activity of the major compounds in FG seeds, such as linoleic acid, palmitic acid, pinene, 4-Pentyl-1-(4-propylcyclohexyl)-1-cyclohexene and linoleic acid methyl ester [10,17,20]. FG is also being studied for its cardiovascular benefits [18].

In particular interest, previous studies including our own (Table 1) showed that FG plants can be a good source of essential minerals, such as Mg^+2^, Zn^+2^, Ca^+2^, PO_4_^3−^, K^+^, Na^+^ and Fe^2+^ [17,18]. Therefore, the incorporation of such ions into the synthesized bone minerals can have a favorable impact on fast bone formation. On the other hand, it is speculated that, by increasing the concentration of the hydroxyl groups in the precursor solution, the interaction between the organic molecules existing in the FG extract and the metallic ions of the calcium phosphate is increased, which means higher interference by apatite crystals during the biosynthesis process. This phenomenon might lead to denser structures being formed by FG organic molecules of the BAp minerals. A denser structure leads to smaller interplanar spacings in the obtained bone minerals. Accordingly, the crystallite size can be reduced to its smallest size to obtain apatite with 2D fine plate-like structures. Controlling the structure of apatite crystals at the nano-level is vital for acquiring a favorable commercial product. In this work, BAp minerals with a 2D plate-like morphology mimicking the natural bones’ composition and structural features, fabricated by biosynthesis of the plant extraction method, have been prepared successfully. The plant polyphenolics possess high cognation for metal ions because of the existence of the hydroxyl groups of phenolic compounds and their molecular structure. As an effort to prepare BAp 2D plate-like structures mimicking the morphology of biological apatite, a biosynthesis process was developed based on the use of natural macromolecules from FG extracts. The bioactive ceramic formed in FG extraction has been called biosynthesized BAp nanoplate, because its composition and structure are similar to those of natural bone rather than of sintered stoichiometric HAp, and it has important characteristics such as low crystallinity and nanoscale sizes that are important for the reabsorption and remodeling found in bone [21,22]. The BAp with the 2D nanostructures formed in this FG extract is believed to exhibit even higher bioactivity and biocompatibility than stoichiometric HAp [22,23]. Furthermore, BAp ultrafine 2D plate-like structures were prepared by employing FG seed extract using biosynthesis wet-chemical precipitation as a simple, efficient, economic and non-toxic route.

## 2. Materials and Methods

### 2.1. Biosynthesis Process

In total, 20 g of commonly used FG seeds were washed several times before being boiled for 15 min [24] in 100 mL Milli-Q water (18.2 MΩ·cm), as typically prepared in traditional medicine. The extracted solution was filtered twice through Whatman no. 1 filter paper. The extract solution was then used for the synthesis of BAp powders by the wet-chemical precipitation route, as described in our previous report [4]. Briefly, calcium cations (1 M Ca(NO_3_)_2_·4H_2_O) (Sigma Aldrich, South Korea) and phosphate anions (0.6 M (NH_4_)_2_HPO_4_) (Sigma Aldrich, South Korea) were separately dissolved in 100% concentration FG extract solution. The phosphate solution was added dropwise at a rate of 0.4 mL min^−1^, under vigorous mixing, into the calcium solution. Ca/P ratio was controlled to be 1.67, the stoichiometric value of HAp. The resultant precipitate slurries were dispersed in a mixing solution of pure Milli-Q water and ethanol (volume ratio = 1:1) and then left to dry in a vacuum for 24 h. The dried powder was further heat-treated at 650 °C for 4 h, with a heating rate of 20 °C/min [4].

The chemical element concentration of the FG seed extract and the biosynthesized powder (0.1 mg dispersed in 5 mL Milli-Q water) were analyzed at 670.783 nm wavelength using a Varian (Inc., Melbourne, Australia) Vista Pro (MPX) radial inductively coupled plasma atomic emission spectroscopy (ICP-OES) instrument. Standards from 0 to 5 mg/L metallic ions were prepared from Fluka, TraceCERT 1000 mg/L stock standard. In addition, due to the limitation of ICP-OES in detecting SO_4_, Cl and CO_3_, the colorimetric analysis method was used to measure the concentration of these elements in both the FG extracts and obtained BAp powders. Electrical conductivity was measured by EC meter CM 40G Ver 1.09 (DKK TOA Co., Tokyo, Japan). The molecular structure of the FG seeds was analyzed using Fourier transform infrared (FTIR) spectroscopy (Prestige-21 FTIR spectrophotometer) spectra in transmission mode.

The morphology of the biosynthesized powders was examined using a Scanning Electron Microscope equipped with Energy Dispersive Spectroscopy (EDS) (SEM, JEOL JSM-6400, Tokyo, Japan) and Transmission Electron Microscopy (TEM, CM 200, Philips, PA, USA). The phase composition, molecular structure and interactions of the obtained powders were analyzed using an X-ray diffractometer (XRD; Rigaku, Tokyo, Japan) and FTIR spectroscopy.

### 2.2. In Vitro Cell Culture

The cell culture was carried out by the indirect extraction method [4]. Briefly, the biosynthesized BAp powders were immersed in 70% (w/v) ethanol for 1h and air-dried for 48 h in a biosafety cabinet class II. Then, the apatite particles were dispersed in Dulbecco’s modified Eagle’s medium (αMEM) supplemented with 10% fetal bovine serum (FBS) in a humidified incubator with 95% relative humidity and 5% CO_2_ at 37 °C, under shaking, for 4 days. Subsequently, the extraction of the previous step was centrifuged for 4 min at 1000 rpm. After incubation, the suspension extracts were used for culturing cells. 10 × 10^3^ MC3T3 cells in 20 µL culture medium were carefully pipetted into the 48 well-plate containing 1 mL extraction. After incubation for 1 and 3 days, live/dead and MTT assays were carried out. The stained scaffolds were further washed two times using Phosphate Buffered Saline (PBS) and observed under a confocal microscope (Nikon ECLIPSE Ti) at 10× magnification. PI stained red-colored dead cells (Exc/Em: 561 nm/570–1000 nm) and FDA stained green colored live cells (Exc/Em: 488 nm/500–550 nm) were observed at 200 µm depth. For SEM observation, at 1 and 3 days, the samples were fixed with 3% Glutaraldehyde in 0.1M Cacodylate buffer and sequentially dehydrated in ethanol (at 25%, 40%, 50%, 80%, 90% and 100%) and treated with hexamethyldisilazane for 1 h. Before SEM observation, the samples were coated to 10 nm thickness using gold sputtering.

## 3. Results and Discussion

To investigate the elements’ concentration, conductivity and chemical constituents at the molecule level of the FG seed extract, ICP-OES, FTIR and solution conductivity were performed. It seems that FG seed extract has an element concentration in this order: K > Cl > CO_3_ > P > Na > SO_4_ > Mg > Ca, and a trace element of Fe and Zn (Table 1). This is in a good agreement with the higher solution electrical conductivity of FG seed extract (1145 µS/m) than that of pure Milli-Q water as a control (93 µS/cm). This indicates that FG seed extract contains soluble metallic ions, resulting in an increased solution conductivity. Wave numbers of 569, 1050, 1250, 1330, 1397, 1545, 1651, 1746, 2956 and 3354 cm^−1^ of FG seeds were detected by the FTIR spectrum analysis (Figure 1). This can be attributed to the existence of organic constituents (such as hydroxyl and carboxyl groups) in FG seeds. The FTIR’s characteristic peaks in the FG seeds were similar to those in the in situ plant cell wall. The distinctive band at 3354 cm^−1^ was due to the vibration of O–H. The presence of water resulted in a weak broad intensity peak in the FG seeds. The peaks located at 2956 and 1397 cm^−1^ are a signature of C–H vibration. The three peaks at 1746, 1397 and 1330 cm^−1^ indicated the presence of the carboxylic group. The existence of -O- was resolved by combining scans showing the peak located at 1050 cm^−1^ with the antisymmetric peak at 1250 cm^−1^ and the symmetric peak at 1050 cm^−1^. The band at 1651 cm^−1^ indicated the presence of the C=C in the material.

The SEM images, shown in Figure 2A,B, show no significant difference between the biosynthesized bone mineral powder’s distribution before and after heat sintering. Furthermore, it can be seen that a uniform distribution of nanoparticles was obtained (Figure 2A,B). To have a close look, the shape and the morphology of biosynthesized BAp powder was further investigated by TEM analysis. TEM results showed that the non-sintered (dried) and sintered (treated at 650 °C) powders had a 2D plate-like nanostructure morphology with an average width (or length) of 12.5 ± 2 and thickness of 3.8 ± 1.2 nm (Figure 2). However, the dried samples had a higher tendency to agglomerate (Figure 2A) than the sintered samples (Figure 2B). Although we have employed the exact chemical precipitation route which has been previously used in several reports, including ours [4,25,26], in the present study, we obtained 2D plate-like nanostructures with a smaller size. This is likely due to the contribution of FG extracts and their organic molecules, which controlled the bone mineral crystals’ growth rate during the biosynthesis process. Formation of 2D plate-like nanostructures may be understood by the concept of driving force playing an important role in the structural changes of a material [4]. Driving force acting during the biosynthesis process of a material generally minimizes the surface area to maintain a smaller surface to volume ratio of the particle, resulting in the lower energy state of the obtained material, which means a more stable structure. Further discussion of the contribution of driving force and controlling the energy state of crystal growth of mineralized nanoparticles was detailed previously [4]. Notably, the morphology of apatite 2D plate-like nanostructures is similar to that of the apatite-like crystals in natural bones, though the plate size of the former is thinner than that of the latter [2,27].

The elemental analysis in mg/L of the obtained bonelike apatite nanoplates is summarized in Table 1. From the obtained results, the obtained 2D nanoplates have most of the elements that are in natural bone. In addition, it is shown clearly that the Ca/P ratio is about 2.93, which is higher than that of stoichiometric HAp fabricated by a similar method. It is likely that the loading of other elements distracts the ratio of Ca/P in the fabricated BAp minerals. The EDS elemental analysis results of the obtained powders are shown in Figure 2C,D. From EDS analysis, the Ca, P and O elements are mainly derived from the Ca–P reagents; the C element is derived from FG extract, as verified by colorimetric analysis (Table 1). In addition, it is obvious that the biosynthesized powders contained ion substitutions, such as Na, Mg and K. The fabricated bone mineral nanoplates showed a high content of potassium ions (822 mg/L), which were derived from FG seed extract (Table 1). Previous epidemiological and clinical studies have demonstrated that potassium ions have a crucial role in regulating the blood pressure of high blood pressure patients and reducing the stroke risk [28], reducing the demineralization of bone (osteoporosis), and reducing the formation of kidney stones [29].

The bone minerals prepared by the biosynthesis method produced a powder of a greenish-beige color [30] (inset of Figure 3A,B). It is likely that the presence of Mg, K and Na compound content in the obtained BAp powder is responsible for producing the color, whereas the appearance of HAp powder obtained by free-FG extract is always white in color [2,27]. These ions are derived from the FG extract solution used as the ion source during the reaction process (Table 1). FG seed extract has been reported to be generally a good source of inorganic minerals [17,18]. The formation of 2D plate-like morphology is therefore likely attributed to the effect of Mg^+^ ions [3,31], which are leached out of FG seeds and incorporated into the biosynthesized bone minerals. Furthermore, previous reports, including ours, have proved that these kinds of substitutions play crucial roles in the biological performance of the bone materials in comparison to pure stoichiometric Hap, and become similar to that of the natural apatite [2]. In particular, Mg plays an important role in the early stages of osteogenesis, stimulating osteoblast growth and differentiation [4,32]. Sodium, available in abundance next to calcium and phosphorus, plays a significant role in bone metabolism and osteoporosis. Fe can be an interesting growth factor molecule related to selective activation of mechanosensitive ion channels using magnetic particles, as reported by Hughes et al. [33].

The final BAp biphasic powder was obtained as illustrated by XRD (JCPDS card: No. 09-0432) analysis (Figure 4A). From X-ray diffractometry profiles, it is revealed that biosynthesis of BAp minerals using FG seed extract is a suitable method to produce bone mineral nanoplates before and after heat sintering. No noticeable diffraction peak appeared, other than that of the calcium phosphate phase structure; however, EDS showed doped trace elements in the fabricated bone minerals. This is likely due to the small amount of these elements that could not be detected by XRD. Thus, the calcium phosphate phase still has its crystal structure and there was no second phase to be detected after heat sintering. XRD was also used to characterize the crystal size. The Scherrer equation (Equation (1)) was applied to calculate the crystal size as follows:(1)$$d=\frac{K\lambda }{B\;cos\theta }$$
where *d* is the average diameter, *K* is the shape factor, *B* is the FWHM of the diffraction peak measured in radios, *λ* is the wavelength of the X-rays and θ is the Bragg’s diffraction angle. The diffraction peak at (0 0 2) plane was chosen for the calculation of crystallite size, since it is sharper and isolated from the others. It was found that the crystal size of biosynthesized 2D nanoplates was 5.75 and 5.53 nm before and after heat sintering, respectively.

Such behavior is in good agreement with the FTIR analysis which was used to follow the evolution of the chemical compounds contained in the initial, and up to the final, BAp plate-like nanostructures. In particular, the FTIR spectrum of the non-sintered and sintered bone mineral nanoplates (Figure 4B) shows peaks typical of carbonated apatite products, and the characteristic bands of carboxylic and carbonate (1400–1600 cm^−1^), phosphate (564, 603, and 900–1100 cm^−1^), amino (≈1405, 1600 and 3200 cm^−1^), hydroxide (630 and 3560 cm^−1^) and acetate (≈2810, 2307 cm^−1^) groups are distinguishable [27]. Together with the strong bands of the CO_3_^2−^ group at ≈1540 cm^−1^, this suggests that it is carbonated apatite [27] as expected, since the precipitation process involves organic reagents from FG extracts [25,26]. Therefore, the phase purity of the final biosynthesized powder in terms of chemical substitutions (HPO_4_^2−^ and CO_3_^2−^ can be present as partially substituting groups of PO_4_^3−^ and/or OH^−^ in the carbonated apatite structure) have been confirmed by the FTIR analysis. The existence of carbonate ions has remarkable advantages for bone mineral crystals in terms of excellent biocompatibility and high resorbability, which make it one of the best candidate materials as a bioresorbable bone substitute [27]. Furthermore, the addition of carbonate in the fabricated bioactive minerals affects the charge balance and crystal structure of the BAp [1,4].

The following mechanism of the formation of BAp 2D plate-like nanostructures can be proposed: the polyphenolic OH^−^ groups of FG extract produced a p-track conjugation effect and bound with Ca^2+^ ions to form Ca^−^ complex by the conjugation effect [17]. During the addition of (NH_4_)_2_HPO_4_ to the Ca(NO_3_)_2_·4H_2_O) while both of them were dissolved in the FG extract solution dropwise, the negative PO_4_^3−^ of the group had an ionic interaction with positive Ca^2+^ ions for the formation of calcium phosphate–complex. Once the concentration of the Ca^2+^ and PO_4_^3−^ reached super saturation after the increase in the ion-release amount, bone mineral nucleates assembled into the nanoplates’ structure with the growth in anisotropic characteristics of a, b and c- planes of apatite crystals. The greater concentration of reactants increased the saturation, resulting in the acceleration of the nucleation of nuclei formation than that of their growth. It is widely accepted that organic molecules adjust many aspects of BAp formation, including control of the phase, shape, orientation and organization of mineral deposits in mineralized tissues. The FG seed extract possesses high flavonoids and other natural bioactive products such as lignin, saponin and vitamins [11]. The reduction of chloroauric acid by using the powerful reducing agents in FG seed extract acts as a better eco-friendly surfactant [18,34,35]. The carboxylic (COO^−^) group, CN and CC metabolites functional groups exist in the seed extract. The metabolites might act as a surfactant of the crystals and the flavonoids might stabilize the electrostatic stabilization of plate-like nanostructures [11,19]. Thus, the seed extract acts as a soft template and controls the growth formation of the BAp plate-like fine structure. The metabolites’ interaction can form relatively tight coverage on the apatite plate-like surface to improve the surface’s ordered structure and crystallinity. Finally, the resulting product was heat-treated at a higher temperature and the impure organics should be eliminated to get higher crystalline apatite phase structure.

Figure 5A–C presents the live/dead cytotoxicity assay results of the viability of the cells on the in vitro cell culture. The results indicate that the sintered BAp mineral 2D plate-like samples showed good viability over 1 and 3 days of culture, consistent with previous reports [36,37], but with no significant difference compared to the control group (one way ANOVA, *p* > 0.05). The quantification results of the MTT proliferation assay to investigate the metabolic activity of the osteoblast cells after 1 and 3 days of culture are shown in Figure 6A. There was a statistical difference in the cell growth between the control (tissue culture plate) and sintered samples. An increase in cell growth was experienced between days 1 and 3 for sintered samples. This growth may indicate the active proliferation of the cells on the mineralized samples and the areas of the samples where calcification is denser. SEM images (Figure 6B,C) of the osteoblast cells showed heather morphology and higher growth on BAp mineral 2D plate-like control samples with no significant difference, *p* > 0.05. Based on the obtained results, BAp displayed the desirable properties for developing bioceramic bone implants.

## 4. Conclusions

Our goal in the present study was to synthesize carbonated bonelike apatite 2D plate-like nanostructures in the presence of fenugreek seed extract aqueous solution as a simple, eco-friendly and economic method. The method was successful in producing bone mineral 2D nanoplates very similar to the chemical composition and structure of natural biological apatite. The TEM analysis showed the presence of well-dispersed 2D plate-like structures in the size range of 12.67 nm, which aligned with the existence of FG organic molecules. Therefore, as anticipated, increases in the organic molecules’ concentration can manipulate the role of hydroxyl pendant groups of FG extracts as a nucleation site for apatite crystals. Hence, more of the OH^−^ group enhances the formation of bone mineral 2D nanoplates in the reaction process, and it will hinder the absorption of Ca^2+^ on hydroxyl pendant groups located in the FG seeds extract. The formation of BAp nanoplates would be more sluggish while there is more intimate interaction between the organic molecules and ionic phases during the reaction. Thus, for the best interaction between both compartments while at the same time allowing the formation of BAp 2D nanoplates within the structure at a higher concentration of hydroxyl groups. The bone mineral 2D nanoplates synthesized by this green precipitation mediated method, with high potassium content besides other useful minerals, can be used as good biomaterials for various biomedical applications. The as-synthesized BAp nanoplates can also be used as a coating material or a nanofiller for orthopedic applications and further studies will be published later.

## Figures and Tables

**Figure 1 nanomaterials-10-00919-f001:**
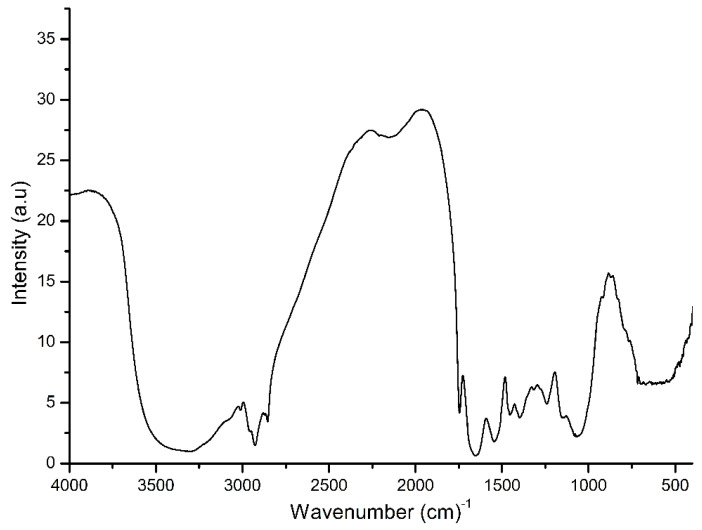
FTIR spectrum in transmission mode of fenugreek seeds.

**Figure 2 nanomaterials-10-00919-f002:**
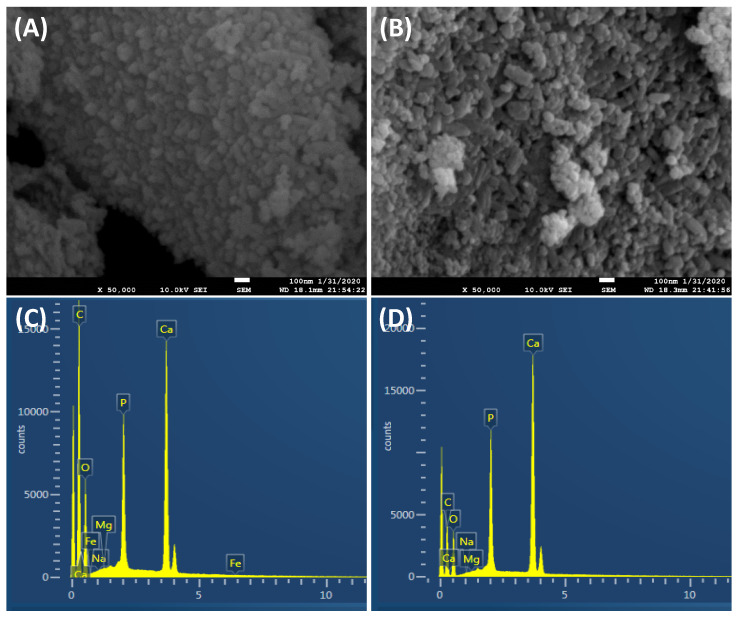
SEM (**A**,**B**) and EDS (**C**,**D**) of the biosynthesized BAp powder by employing FG seed extract. (**A**) and (**C**) are non-sintered, and (**B**) and (**D**) are sintered powder samples.

**Figure 3 nanomaterials-10-00919-f003:**
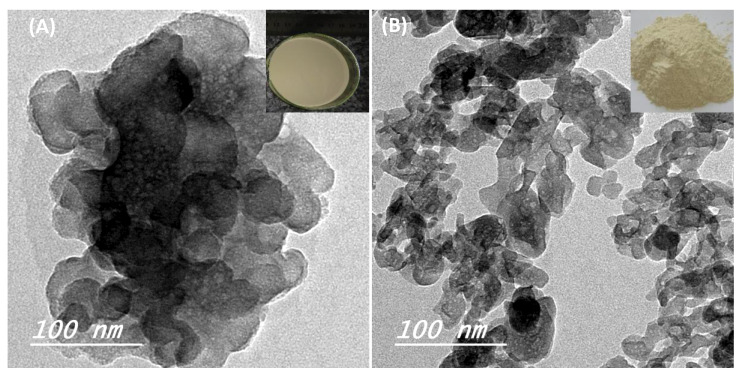
TEM morphology of non-sintered (dried) (**A**) and sintered (heat treated at 650 °C) bone mineral powders (**B**). Inset 2D images show the greenish-beige color of the apatite powders obtained by the biosynthesis method.

**Figure 4 nanomaterials-10-00919-f004:**
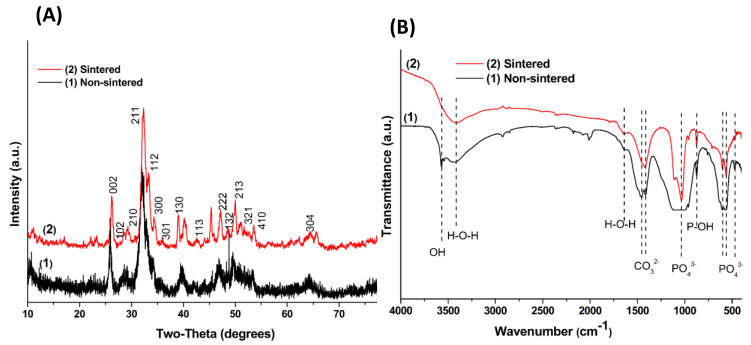
XRD (**A**) and FTIR (**B**) profiles of the fabricated HAp 2D plate-like nanostructure using FG extract.

**Figure 5 nanomaterials-10-00919-f005:**
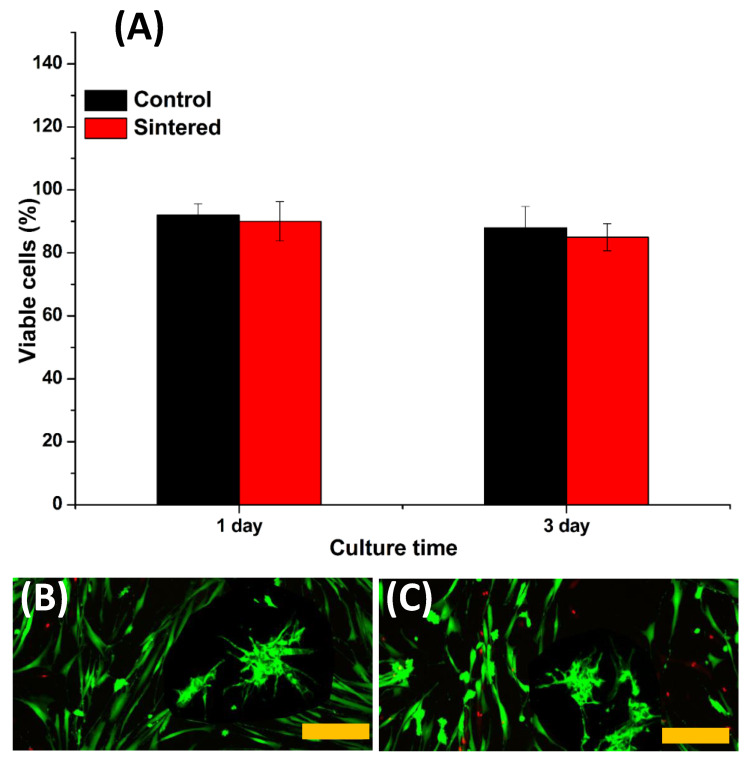
(**A**) Live /dead viability assay; (**B**,**C**) laser confocal images of MC3T3 osteoblast-like cells after 3 days of culture. Cytotoxic events that affect cell membrane integrity can be accurately assessed using this method. The live and dead cells exhibited green and red fluorescence, respectively. Dead cells pretreated with 70% ethanol for 30 min were used as controls for dead cells. The scale bar represents 100 µm.

**Figure 6 nanomaterials-10-00919-f006:**
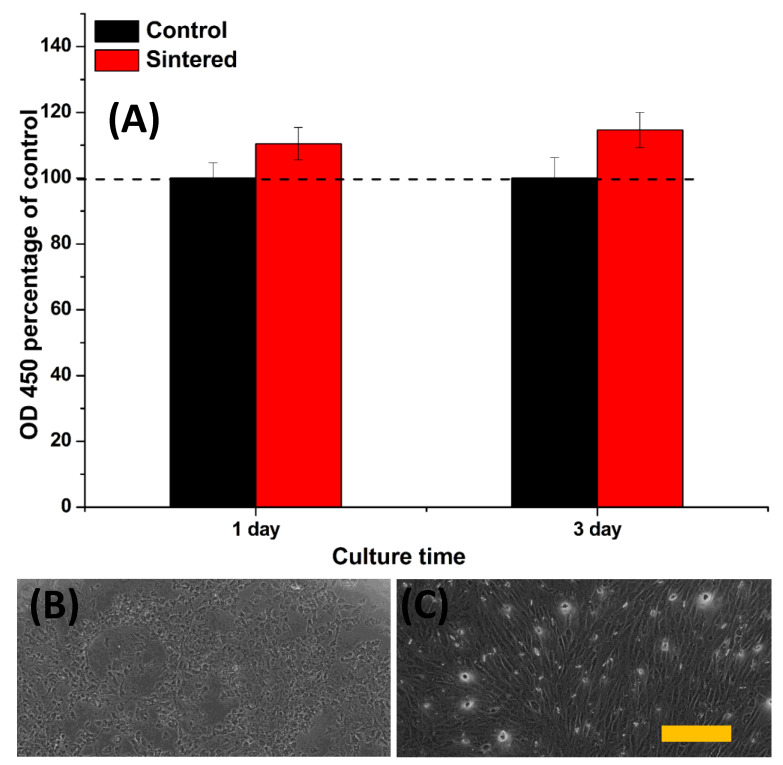
(**A**) MTT proliferation assay and (**B**,**C**) SEM images of MC3T3 osteoblast-like cells. The scale bar represents 100 µm.

**Table 1 nanomaterials-10-00919-t001:** Chemical analysis of fenugreek seed extract and biosynthesized BAp powders in mg/L.

Samples	Ca	Fe	K	Mg	Na	P	Zn	SO_4_	Cl	CO_3_
FG seeds extracts	17	0.39	751	20	32.2	35	0.14	32	83.7	56
BAp minerals	2978	0.45	822	32	39.7	1014	0.24	38	78.5	169

Samples were analysed by ICP-OES, whereas, other results were analysed calorimetrically for SO_4_, Cl and CO_3._

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
