# Peer review of "Biosynthesis of Bonelike Apatite 2D Nanoplate Structures Using Fenugreek Seed Extract"

_nanomaterials, 2020, doi:10.3390/nano10050919_

Round 1
Reviewer 1 Report
I think that all my observations have been properly considered. In my opinion the paper can be published in the present form
Author Response
R1: Thank you for your positive feedback.
Comments and Suggestions for Authors
I think that all my observations have been properly considered. In my opinion, the paper can be published in the present form
R1: Thank you for your positive feedback.
Reviewer 2 Report
The authors significantly addressed the concerns presented by the reviewer. The manuscript is in much better shape compared to the original version.
However, there still are a lot of small mistakes (spelling, grammar, editing etc.). Just to prove my points, I am listing the errors found in the first page.
- line 17, "A" needs to be deleted
- line 25, BAp should be named as HAp BAp if the authors want to use this name uniformly throughout the manuscript
- line 41, Mg+2 should be Mg2+
More important feedback is on the Figure 6 (the in vitro cell culture data). As the author argues, it is correct that the effects of conventional HAp on the growth of the osteoblast has been extensively studied. Ironically more so, it is meaningless to say anything about the unique aspect (bio-efficacy) of the nanocrystals made by the current study, if the efficacy of the nanocrystals made by the proposed method is directly compared to those of conventional nanocrystals.
At the least modification, the difference in viability presented in the graph (Figure 6) should be statistically tested (ex. p-values from student t-test), presented, and discussed.
Author Response
- Thank you for the feedback. All comments were well-considered accordingly.
Comments and Suggestions for Authors
The authors significantly addressed the concerns presented by the reviewer. The manuscript is in much better shape compared to the original version.
- Thank you for the positive feedback.
However, there still are a lot of small mistakes (spelling, grammar, editing, etc.). Just to prove my points, I am listing the errors found in the first page.
line 17, "A" needs to be deleted
line 25, BAp should be named as HAp BAp if the authors want to use this name uniformly throughout the manuscript
- Thank you for the positive feedback. In the present study, HAp terminology was used to refer to stoichiometric hydroxyapatite, whereas BAp was used for Bone like apatite minerals
line 41, Mg+2 should be Mg2+
- the whole manuscript has been polished and corrected, accordingly.
More important feedback is in Figure 6 (the in vitro cell culture data). As the author argues, it is correct that the effects of conventional HAp on the growth of the osteoblast has been extensively studied. Ironically more so, it is meaningless to say anything about the unique aspect (bio-efficacy) of the nanocrystals made by the current study, if the efficacy of the nanocrystals made by the proposed method is directly compared to those of conventional nanocrystals.
- I definitely agree with the reviewer, further biological studies are essential to prove that the new developed bioactive ceramic materials have higher bioactivity compared to the conventional bioceramics. The main aim of the present study was to fabricate and characterize the bioactive materials and also to investigate its cell viability. Our future study will be focused on the absorption properties and biological functionality.
At the least modification, the difference in viability presented in the graph (Figure 6) should be statistically tested (ex. p-values from student t-test), presented, and discussed.
- the P-value was calculated and it is shown in the manuscript (line 294). The results did not show significantly different (one way ANOVA, P>0.05) between the groups. I fully agree with the reviewer, further in vitro experimental such as gene expressions, osteoblasts cell differentiation are needed to further prove the bioactivity of the BAp bioceramics materials.
Round 2
Reviewer 2 Report
The authors diligently addressed the concerns presented by the reviewer. The manuscript is in much better shape compared to the previous version.
This manuscript is a resubmission of an earlier submission. The following is a list of the peer review reports and author responses from that submission.
Round 1
Reviewer 1 Report
This paper describes a process for synthesis of hydroxyapatite using Fenugreek seed extract.
Even if the process can be novel in a sense that the particular plant extract is newly employed for the synthesis of hydroxyapatite, the paper is not very appealing because of the lack of rational behind why Fenugreek seed extract could be the best candidate.
In some parts of results and discussion, very interesting thoughts on how the components of the seed extract contribute to the composition and structure of the final hydroxyapatite crystals. However, in order to scientifically discuss these aspects, the composition of the employed seed extract needs to be characterized and presented first.
The current form of the paper is written in haste. The grammars of so many sentences are wrong, and many sentences are not closing, and such sentences are appearing even in the abstract. There are missing sub-section titles (for example, there are only 2.1 and 2.2 is missing and later 1.1 appears).
The following discussion and experimental designs should be revised/addressed.
- In page 4, line 130-133, how a driving force to minimize the surface area can induce in this case a smaller-sized crystal? The rationale needs more elaborate explanation.
2. In page 6, line 185-186, the authors are mentioning the effect of carbonate in HAp on the crystal structure? How does it appear in the current XRD data? It needs to be fully addressed.
3. The discussion in page 7 would be a very significant discussion only if the manuscript shows that the contents (%) of the corresponding components in their seed extract.
4. Page 9, line 226, is it P>0.05?
5. The paragraph in Page 10, line 240-253, is not described in the Experimental section. The characterization of the resulting 3D scaffold is very preliminary. If the authors believe the proposed method would promote the regeneration of bone tissues, at least some in-vitro experiment needs to be added.
6. The results shown in Figure 4 and 5 need a positive control group. The authors included a tissue culture plate as a negative control group, but the experimental group needs to be compared to a proper positive control group as well. The proper positive control group could be a hydroxyapitite nanocrystals prepared by a conventional nano-precipitation, to demonstrate the real potential benefit of the presented synthetic route.
Reviewer 2 Report
The paper is interesting as it concerns a particle engineering study to obtain HAp nanoparticles with specific shape.
I think however that the method section should be improved:
The preparation method of HAp powders is not too clear. The authors refer to an emulsion-precipitation method. It is not clear when an emulsion is obtained. “Vigorously mixed” how? What’s the meaning of 0.4 ml/min rate? Is the way of mixing relevant?
What is the composition of FG extract? Polyphenolic groups are claimed as relevant for the nanostructures obtained: what are the components of FG extract involved in this effect? What was their concentration and how can it be controlled? The same for the effect of Mg ions.
The preparation of fibers is not described in the methods. Although references are given, some information must be given here. The apparatus, the experimental conditions, the formulative details (PCL in which solvent).
It should be useful to show a comparison with nanoprecipitates obtained with the same method without FG extract (a control sample)
Some English revision would be useful.